# Pathophysiological Roles of the TRPV4 Channel in the Heart

**DOI:** 10.3390/cells12121654

**Published:** 2023-06-17

**Authors:** Sébastien Chaigne, Solène Barbeau, Thomas Ducret, Romain Guinamard, David Benoist

**Affiliations:** 1IHU LIRYC Electrophysiology and Heart Modeling Institute, Fondation Bordeaux Université, 33600 Bordeaux, France; 2Centre de Recherche Cardio-Thoracique de Bordeaux, INSERM U1045, University of Bordeaux, 33600 Pessac, France; 3Electrophysiology and Ablation Unit, Bordeaux University Hospital, 33604 Pessac, France; 4UR4650, Physiopathologie et Stratégies d’Imagerie du Remodelage Cardiovasculaire, GIP Cyceron, Université de Caen Normandie, 14032 Caen, France

**Keywords:** TRPV4 channels, TRPV4-KO mice, heart, myocytes, fibroblasts, endothelial cells, action potential, calcium homeostasis

## Abstract

The transient receptor potential vanilloid 4 (TRPV4) channel is a non-selective cation channel that is mostly permeable to calcium (Ca^2+^), which participates in intracellular Ca^2+^ handling in cardiac cells. It is widely expressed through the body and is activated by a large spectrum of physicochemical stimuli, conferring it a role in a variety of sensorial and physiological functions. Within the cardiovascular system, TRPV4 expression is reported in cardiomyocytes, endothelial cells (ECs) and smooth muscle cells (SMCs), where it modulates mitochondrial activity, Ca^2+^ homeostasis, cardiomyocytes electrical activity and contractility, cardiac embryonic development and fibroblast proliferation, as well as vascular permeability, dilatation and constriction. On the other hand, TRPV4 channels participate in several cardiac pathological processes such as the development of cardiac fibrosis, hypertrophy, ischemia–reperfusion injuries, heart failure, myocardial infarction and arrhythmia. In this manuscript, we provide an overview of TRPV4 channel implications in cardiac physiology and discuss the potential of the TRPV4 channel as a therapeutic target against cardiovascular diseases.

## 1. Introduction

The mammalian TRP channels are widely expressed in the heart and can be considered as real “cellular switches” able to respond to numerous physical and chemical stimuli related to sensory physiology [1,2,3]. In mammals, the TRP family is composed of 28 members and classified according to amino acid sequence homology into six families: TRPA (ankyrin; TRPA1), TRPC (canonical; TRPC1–TRPC7), TRPM (melastatin; TRPM1–TRPM8), TRPML (mucolipin; TRPML1–TRPML3), TRPP (polycystin; TRPP1–TRPP3) and TRPV (vanilloid; TRPV1–TRPV6) [3,4]. With some exceptions, most of the TRP channels are able to conduct Ca^2+^, which plays an important role in stimulus–response reactions of cardiac cells. Interestingly, the altered expression, localization and regulation of TRP channels have already been related to cardiovascular disorders due to Ca^2+^ handling dysregulation [4]. Among the TRPV channel family [5], the TRPV4 channel has emerged as a key modulator in cardiac cell structure and activity [6,7]. The recent identification of pharmacological modulators and construction of TRPV4 knockout (KO) mice unmasked several physiopathological roles of this channel [5,8,9]. However, the mechanisms that regulate cardiac TRPV4 channels still remain poorly understood. In this context, this review will discuss the latest findings on the pathophysiological role of the TRPV4 channel in the cardiovascular system and why an understanding of TRPV4 regulation may lead to novel therapeutical strategies related to cardiac diseases.

## 2. Gene, Structure, Function and Electrical Properties 

The TRPV4 channel has received particular attention due to its large expression in the cardiovascular system [5,8,10,11]. The TRPV4 protein is encoded by the *trpv4* gene, present on the long (q) arm of chromosome 12 at position 24.1 in humans [4]. The corresponding locus is found between bases 109,783,087 and 109,783,406 of the genome and is composed of 15 exons. Interestingly, five channel isoforms (TRPV4-A to -E) were found and produced by alternate exon splicing in a human bronchial epithelial cell line. Only the TRPV4-A and TRPV4-D isoforms are localized in the plasma membrane and display identical biophysical properties [12], while the other splice variants (N-terminal deletions) remain in the endoplasmic reticulum (ER) and do not form functional ion channels [12]. Because the functional differences between these two isoforms have not been explored in the cardiovascular system, we will refer in the following section to TRPV4 as the TRPV4-A and TRPV4-D isoforms without distinction. TRPV4 has a tetrameric structure, with each subunit being composed of six transmembrane segments (S1 to S6), a pattern shared with other TRP channels and voltage-gated ion channel subunits (VGIC). The S1 to S4 segments constitute the peripheral structure, while the central S5-S6 loops border the pore of the channel [13,14,15,16]. The N- and C-terminal extremities are intracellular and contain a variety of functional domains [17]. A recent crystallographic study combined with an X-ray approach gives more details about the structure of *Xenopus tropicalis* TRPV4, with a resolution of 3.8 Å [16]. 

The N-terminal region represents more than half of the protein and plays a critical role in channel assembly, trafficking and regulation [16,18,19,20,21,22]. It harbours six Ankyrin repeated domains (ARD1–6) that participate in channel oligomerization, protein–protein interaction and trafficking [4]. Note that the absence of these specific domains blocks the TRPV4 trafficking at the ER level [12]. This region also presents a Ca^2+^-calmodulin kinase type II (CaMKII) regulation site conferring channel sensitivity to intracellular Ca^2+^. In HEK-293 cells overexpressing TRPV4, a proline-rich sequence in the TRPV4 N-terminal can interact with the cytoskeleton protein PACSIN 3 (protein kinase C and casein kinase substrate in neurons 3), thereby regulating channel trafficking and preventing or reducing TRPV4 activation by heat, cell swelling and arachidonic acid [23]. In this context, PACSIN3 can be considered as a TRPV4 channel regulatory protein that regulates both the TRPV4 subcellular localization and its function. Obviously, further studies on cardiomyocytes are needed to better understand the mechanism and the role of this interaction. In addition, previous studies have shown on several heterologous expression systems involving HEK-293, Hela and SJSA-1 cells overexpressing TRPV4 that the TRPV4 N-terminal region is able to interact with human OS-9, an ER-resident lectin, in order to prevent channel trafficking to the plasma membrane [24]. This point is relevant because it suggests that OS-9 is also an important auxiliary protein for TRPV4 maturation.

The human TRPV4 C-terminal extremity contains different domains: (i) “TRPbox” (carrying the consensus sequence WKFQR) [25]; (ii) a CaM-binding domain that modulates its tetramerization and its gating in both HEK-293 and T-REX cells [26], as well as Ca^2+^ influx into oocytes [27]; (iii) a PDZ domain, which interacts with numerous cell auxiliary proteins [4,13]. Furthermore, the microtubule-associated protein 7 (MAP7) interaction with the C-terminal extremity of TRPV4 increases both expression and functional activity in the plasma membrane in Chinese hamster ovary cells [28]. Because MAP7 is directly related to the cytoskeletal filaments, this interaction may underlie the TRPV4 mechanotransduction mechanism [28]; nevertheless, this hypothesis requires further attention in myocytes.

TRPV4 principally assemble as homotetramers [13,29] to form a functional channel but can also form heterotetramers with other members of the TRP channel family, namely TRPC1 [30,31] and TRPP2 [29,31]. It is important to note that these kinds of biophysical studies were conducted only in classical heterologous expression systems (HEK-293 and tsa 201 cells overexpressing TRPV4) and in native endothelial cells (ECs) (human umbilical vein ECs (HUVECs), mouse primary aortic ECs and rat mesenteric artery ECs (MAECs)), undeniably requiring further research in cardiomyocytes to better understand the heteromeric TRPV4 contribution to cardiac cell function. The TRPV4 channel can also interact with the α-subunit of a few ion channels and aquaporins [32,33,34,35,36,37]. This heteromerization was shown to modify the TRPV4 channel’s biophysical properties in HEK-293 and MAECs cells [31,38,39]. The TRPV4 channel shows a lower voltage sensitivity compared to other TRP channels but still harbors an outward rectification current when expressed in heterologous expression systems, constituting the signature of most TRP channels. To date, this weak voltage dependence of TRPV4 is not yet clearly understood. Nevertheless, the low density of positive charges in the S4 voltage sensor domain [40] combined with a better understanding of its crystal structure [16] provides a plausible mechanism for its gating. Indeed, TRPV4 adopts a domain-swapped arrangement between the S1–S4 domain and the pore domain S5–S6, similarly to TRPV1, TRPV2 and numerous VGICs, whereas the connector between these last two domains adopts an ordered loop structure rather than the α-helix segment present within TRPV1 and TRPV2. This last structural aspect is important, since the VGIC connector acts as a mechanical lever to couple the pore opening and the voltage sensor activation [41,42], and its absence can change the gating behavior [43,44]. Taken together, the interface behavior between the S1–S4 and pore domains in TRPV4 is unique among TRP channels and not closely related to the VGICs. In HEK-293 and native mouse aorta ECs, the TRPV4 single channel conductance rates are ~30–60 pS and ~80–100 pS for inward and outward currents, respectively [45]. Like most of the TRP channels, TRPV4 is mainly permeable to Ca^2+^ over other ions, and its permeability sequence was established to be as follows: Ca^2+^ >> Mg^2+^ > K^+^ > Na^+^ (P_Ca_/P_Na_ ~ 10 and P_Mg_/P_Na_ ~ 2–3) [30,46,47]. Surprisingly, the macroscopic TRPV4 current has not yet been unmasked in cardiomyocytes. A recent structural characterization study performed by Yuan’s group reported a larger selectivity filter (10.6–12.6 Å) within the upper gate of TRPV4 compared to other TRPV channels (TRPV1, TRPV2 and TRPV6, ranging from 4.2 to 7.4 Å) (Figure 1 [16]). 

Note that the presence of this specificity does not significantly increases the degree of Ca^2+^ permeability within this ion channel family but should potentially be considered as an important structural feature when TRPV4 has to deal with mechanical forces associated with increased membrane tension. Moreover, this structural specificity opens up the possibility of developing a new generation of drugs that selectively target TRPV4 channels without interfering with other cardiac ion channels from the same family or not. It may, thus, provide a more targeted approach to treat human arrhythmias and minimize off-target effects.

## 3. Available Tools to Investigate TRPV4’s Roles

The intracellular Ca^2+^ concentration plays a pivotal role in living organisms because it is involved in several physiological regulation processes but also in responses to various pathological states [4,48]. In the myocardium, cellular Ca^2+^ homeostasis maintains normal heart function, which requires various specialized proteins such as ion channels and exchangers [49]. The central role of Ca^2+^ in cardiac excitation–contraction is well-established, where Ca^2+^ is at the interface of the electrical membrane activity (action potential (AP)) and cell contraction [50]. Any disturbance in intracellular Ca^2+^ homeostasis may, thus, lead to contractile or electrical defects in the myocardium. In this context, the TRPV4 function related to Ca^2+^ permeability is evaluated with the greatest attention [51,52]. Consequently, synthetic specific antagonists of TRPV4 have been designed and evaluated to treat human diseases including cardiac pathologies [48,52,53].

### 3.1. Pharmacological Modulators

To date, several synthetic molecules that modulate TRPV4 channels have been developed to aim at treating diseases such as osteoarthritis, respiratory diseases, cancers, gastrointestinal disorders and pain [52]. Recently, some of these compounds entered safety trials to treat heart failure [52]. Progress in the development of these agents led to the last generation of more potent and specific modulators that have been used to unmask the TRPV4 channel’s pathophysiological roles in the heart.

#### 3.1.1. TRPV4 Agonists 

Four generations of TRPV4 agonists (see Table 1) have been used on different tissues and cell types to determine the physiological and pathological functions in the myocardium. The 4α-phorbol 12,13-didecanoate (4α-PDD, a semisynthetic derivative of diterpenoid phorbol), 5,6 epoxyeicosatrienoicacid (5,6-EET, an oxidative metabolite of arachidonic acid), RN-1747 (a benzenesulfonamide) and GSK1016790A agonists (fully synthetic) have been extensively used in cardiovascular investigations [8,11,54,55,56,57]. In addition, Atobe and al. have recently reported on the new TRPV4 agonist quinazolin-4(3 H)-one, a derivative intended to treat osteoarthritis [58] (see Table 1).

#### 3.1.2. TRPV4 Antagonists

Over the last decade, several TRPV4 antagonists were used to highlight TRPV4’s roles in the cardiovascular system. The initial studies used several non-selective TRPV4 antagonists such as ruthenium red (RuR) and gadolinium [59]. Subsequent studies have evaluated the effects of more selective antagonists, including RN-1734, which completely block both ligand- and hypotonicity-activated TRPV4 channels in T-REx and HEK-293 cells without affecting other TRP channels such as TRPV1, TRPV3 and TRPM8 channels [60], or the widely-used HC067047 antagonist, which was used in vivo and in vitro in the context of ischemia–reperfusion [55,61]. Recently, a promising clinical antagonist GSK2193874, which possesses high TRPV4 affinity, has been shown to efficiently prevent and treat lung edema in heart failure models and congestive heart failure [62] (Table 2). Several optimized antagonists exhibiting better in vivo availability (RN-9893) [63], solubility (GSK3527497) [64] or efficacy (novel 2′,4′-dimethyl-[4,5′-bithiazol]-2-yl amino derivatives [65,66] and GSK2798745 [52]) or with decreased toxicity (sulfone pyrrolidine sulfonamide) [53,67] have been developed since then. Knowing that the TRPV4 antagonist GSK2798745 [52] was the first inhibitor tested in humans (phase 2), more homologs are likely to appear in the near future (see below Table 2). Indeed, the recently unveiled Cryo-EM structure [16] of TRPV4 channels combined with the pharmaceutical industry’s efforts should accelerate the knowledge related to the mechanism of action of current modulators and may help to identify the next generation of TRPV4 drugs. 

**Table 1 cells-12-01654-t001:** Overview of the evolution of TRPV4 agonist, the associated physiological processes and their clinical application. Note: 4α-PDD: 4alpha-phorbol 12,13-didecanoate; PKC: protein kinase C; CF: cardiofibroblasts; 5,6-EET: 5,6 epoxyeicosatrienoicacid; mPTP: mitochondrial permeability transition pore; iv: intravenous injection; TNF-α: tumor necrosis factor.

Molecules	Year of Identification	EC50/IC50	Other Targets	Features	Cardiovascular Effects	Clinical Trials/Uses	References
Agonists							
4α-PDD	2003	50 µM	Dorsal root ganglia neurons independently of TRPV4	Negative control for phorbol esters (PKC inhibitors)	Ca^2+^ influx in CF and myocytes Ca^2+^ entry in pulmonary artery smooth muscle cells and increased isometric tension in artery rings	None	[68,69,70,71]
5,6-EET	2003	0.13 µM	mPTP	Metabolite of arachidonic acid by cytochrome P450	Reduction in: vascular tone, inflammatory response, pathological cardiac remodeling (fibrosis, hypertrophy) and apoptosis Improvement in cardiomyocytes function Cardioprotection Promotes angiogenesis	None because of poor solubility and short half life	[72,73,74,75]
RN-1747	2009	5.9–7.7 µM	TRPM8 antagonist (IC50 = 4 µM)	Benzenesulfonamide derivative	None reported	None	[60]
GSK1016790A	2008	1–18 nM	Unknown	Oral administration iv	Endothelial failure and circulatory collapse Reduction in TNF-α induced monocyte adhesion to human endothelial cells and atherosclerosis Cationic non-selective current activation in rat atrial fibroblasts Ca^2+^ influx in CF and differentiation into myofibroblasts and cardiomyocytes Worsening of ischemia–reperfusion injuries in isolated mouse hearts and in H9c2 cell line and neonatal rat myocytes Decrease in systemic arterial pressure, small decrease in pulmonary arterial pressure and small increase in cardiac output	None	[8,51,55,76,77,78,79,80,81,82]
Quinazolin-4(3H)	2019	280 nM	Unknown	Orally bioactive ?	None reported	None in cardiovascular diseases	[58]

**Table 2 cells-12-01654-t002:** Overview of the evolution of TRPV4 antagonists, the associated physiological processes and their clinical application. Note: mPTP: mitochondrial permeability transition pore; iv: intravenous injection; RISK: reperfusion injury salvage kinase; DCM-hiPSC-CMs: dilated cardiomyopathy-induced pluripotent stem cell.

Molecules	Year of Identification	EC50/IC50	Other Targets	Features	Cardiovascular Effects	Clinical Trials/Uses	References
Antagonists							
RN-1734	2009	2 to 6 µM	Poor pharmacokinetics and toxicity	Highly selective	Prevention of Ca^2+^-entry-mediated vasorelaxation of mesenteric arteries Abolition of stretch-activated Ca^2+^ entry in human-induced pluripotent stem cell-derived cardiomyocytes Inhibition of the phenylephrine-induced contraction in pulmonary artery smooth muscle cells but non-specific off-target effects	None	[60,83,84,85,86,87]
HC-067047	2010	17 to 133 nM	ROS production, depolarization of mitochondrial membrane potential (Δψm) and mPTP opening during H/R	IV administration Intraperitonealy injected Potent Selective?	Cardioprotection (significantly reduced infarct size, decreased troponin T levels and improved cardiac function in murine model myocardial I/R) [57] Anti-apoptotic effects via the activation of RISK pathway Reduced TRPV4-related mechanosensitive Ca^2+^ signaling in DCM-hiPSC-CMs Prevents entry of divalent cation in response to myocyte-stretching- and hypoosmotic-stress-induced cardiomyocyte death and ischemia–reperfusion-induced cardiac damage Significantly reduced diabetes-induced cardiac fibrosis Inhibition of the PE-induced contraction in pulmonary artery smooth muscle cells	None	[11,55,62,86,87,88,89,90]
RN-9893	2015	320 to 660 nM	Exhibits > 15-fold selectivity for TRPV4 over TRPV1 and -V3 and TRPM8	Moderate oral bioavailability Potent Selective	Cardioprotection (blocked collagen production following stretching in human valve interstitial cells) Reduced cardiac fibrosis	None	[63,82,88,89,91]
GSK2193874	2017	2 to 50 nM	Unknown	Orally active Potent Selective	Abolition of pulmonary edema associated with heart failure and enhanced arterial oxygenation Increased tail blood flow	None	[76,87,92,93]
GSK3527497	2019	12 nM	Unknown	Suitable for oral and iv administration Reduced bioavailability Poor pharmacokinetics and low solubility	Unknown	None	[64,88]
GSK2798745	2019	2 to 16 nM	Without any clinically significant safety concerns	Highly potent Selective Orally active	Resolves pulmonary edema in heart failure models and attenuates lung damage induced by chemical agents	Heart failure congestive (October 2016), heart failure (13 April 2016) and respiratory diseases (9 June 2014) Diabetic macular edema (7 September 2020) and cough (5 April 2018): https://clinicaltrials.gov/	[48,52,61,76,87,88,89,94]
GSK3395879	2018	1 nM	IC50 > 10 µM for TRPA1, TRPV1, TRPM2, TRPM4, TRPM8, TRPC3, TRPC4, TRPC5, TRPC6	Orally bioactive Highly potent	Abolition of pulmonary edema associated with heart failure	None	[11,67,91,95]

### 3.2. TRPV4 Knockout Mice

To our knowledge, two transgenic TRPV4 knockout mouse strains were developed from two different groups. Suzuki et al. used the 129/SvJ strain via a cassette insertion mutagenesis of exon 5 [96], whereas Liedtke et al. used the C57bl/6J strain with a Cre-lox-mediated excision of exon 12 [97]. Both mouse strains were viable and fertile [96,97] (see Table 3). 

## 4. Physiological Roles in Cardiovascular System

### 4.1. TRPV4 Expression Profile under Physiological Conditions

In mammals, TRPV4 channel expression is distributed in various organs and tissues, including the blood vessels and heart. RT-PCR, immunostaining and functional recordings have demonstrated that the TRPV4 channel is expressed in cardiac cell types such as atrial and ventricular cardiomyocytes [8,10,11,51,55,61,112,113], human embryonic stem cell-derived cardiomyocytes [114], cardiofibroblasts (CFs) [51,81,90,115,116,117], ECs [84,100,118,119,120,121] and smooth muscle cells [71,84,87,120,122–127 ] (see Table 4). 

Taken together, the TRPV4 channel is present and functional in the plasma membrane of cardiac cells, where it plays important physiological roles.

### 4.2. Modulation of Ventricular Electrical Activity

In healthy mammalian hearts, the action potential (AP) waveform initiates and modulates cardiac contractility. It occurs suddenly and transitorily when the resting membrane potential depolarizes and repolarizes according to the successive opening and closing of several ion channels. A lot of attention is paid to the ventricular repolarization phase in preclinical investigations, as AP prolongation is often associated with an increased risk of potentially lethal arrhythmias. Recent evidence has shown that TRP channel opening produces a depolarizing current, since the net flux of cations is inward under physiological conditions [3]. Therefore, these channels are particularly important in excitable cells, such as cardiomyocytes, where they can both trigger and modulate the AP shape. To date, some TRP channels, such as TRPC3 [129,130], TRPM4 [131,132,133] and TRPM7 [134], have been shown to modulate both the sinus node and cardiac AP. Interestingly, the deletion or mutations of TRPM4 were reported to be associated with several cardiac electrical disorders (Brugada, long QT and progressive cardiac conduction disorders) [135,136,137,138,139] and hypertrophy [140]. Since other TRP channels are expressed in cardiac myocytes they may participate in cardiac electrical activity, and their mutations could lead to inherited cardiac electrical disorders. Thus, it appears necessary to specifically describe the effects of each of these channels, including TRPV4, and to evaluate their physiological roles in the cardiovascular system.

TRPV4 protein expression was detected in mouse [8,55,61] and rat [55] left ventricular myocytes. It shows membrane-specific expression and locations (plasma membrane and t-tubules), which are age-dependent [8,11]. In young mice [8], the TRPV4 agonist GSK1016790A induced a dose-dependent and transient increase in AP duration in *trpv4^+/+^* left ventricular myocytes. In this study, GSK1016790A was proposed to promote the trafficking of the TRPV4 channel to the membrane. This phenomenon was transient because of the subsequent rapid channel endocytosis [8,56,141]. A similar effect of GSK1016790A was also observed on rat atrial myocytes [51]. According to its biophysical properties, TRPV4’s potentiation in a physiological or pathological context would prolong AP. Conversely, the TRPV4 inhibitor GSK2193874 (100 nM) significantly shortened the AP duration in *trpv4^+/+^* left ventricular myocytes. The effects of these pharmacological modulators on the ventricular AP were not observed in *trpv4^-/-^* mice, showing that the effects observed on WT cardiomyocytes were exclusively due to TRPV4 modulation. These data suggest that TRPV4 channels carry an inward current, which remains to be characterized, during the ventricular AP in basal conditions. Computational modeling was used to predict the shape of the TRPV4 current during AP [8]. This model considers the channel permeation and open probability depending on the voltage and subsarcolemmal [Ca^2+^]_i_ that will be sensed by the channel. Since the TRPV4 channel’s open probability increases with depolarization and decreases with [Ca^2+^]_i_ elevation, the model predicted a transient inward current that developed rapidly after the AP upstroke and declined with increases in [Ca^2+^]_i_ and membrane repolarization [8]. The consequence of *trpv4* deletion on the cardiac electrical activity was also evaluated. In vivo electrocardiogram recordings revealed a significantly shortened QT interval in *trpv4^-/-^* mice compared to their wild-type littermates (C57bl/6J strain). In accordance with these data, a 19% reduction in AP duration was found in ventricular myocytes from *trpv4^-/-^* mice compared to *trpv4^+/+^* mice. No change in the properties of the main VGICs participating in AP was observed between *trpv4^+/+^* and *trpv4^-/-^* myocytes, suggesting that QT and AP duration shortening in *trpv4^-/-^* mice is exclusively attributable to *trpv4* deletion [8]. Thus, TRPV4 channels are constitutively active in cardiomyocytes from young mice under basal conditions and modulate ventricular electrophysiology [8]. The contribution is likely to increase with age [11]. Moreover, since TRPV4 is well known to be mechanosensitive [86,91,114], membrane stretching due to contraction–relaxation cycles, especially under conditions of acute increases in ventricular load (e.g., during exercise), may modulate its function. Finally, a recent investigation has shown that the TRPV4 expression level is increased in cardiomyocytes of aged (24–27 months) mice compared with young (3–6 months) mice [11] but TRPV4’s contribution to the prolonged repolarization commonly observed with aging remains to be studied.

Together, these studies suggest that TRPV4 modulates ventricular electrical activity under basal conditions and may, thus, be involved in cardiac arrhythmias in aging or under pathological conditions [11] (see Section 5 for details). 

### 4.3. Modulation of Cardiac Contractility

Muscular contractility is onset by an increase in cytoplasmic Ca^2+^. Therefore, Ca^2+^ handling results from a tight balance between both the Ca^2+^ influx, intracellular store release and uptake and cell extrusion [50]. In a ventricular myocyte, Ca^2+^ influx is mainly mediated by voltage-gated Ca^2+^ channels (Cav1.2), whereas the Ca^2+^ extrusion is ensured by the Na/Ca exchanger (NCX1) and plasma membrane Ca^2+^ pump (PMCA) [142]. Several TRP channels are present and functional and can constitute alternative sources of Ca^2+^ entry. Furthermore, it is well established that impaired Ca^2+^ handling leads to abnormal contractility [140,143,144]. Therefore, a better understanding of the TRP channels involved in Ca^2+^ handling and contractility will constitute an important step to treat deleterious cardiac diseases involving cytosolic Ca^2+^ overload in myocytes [145]. Regarding TRPV4, several studies reported its involvement in these processes, while others did not observe an obvious contribution. Indeed, its contribution to intracellular Ca^2+^ influx and myocardial contractility was examined by Li et al., using the TRPV4 agonist 4-αPDD on isolated rat papillary muscles. The authors revealed that the 4-αPDD had no effect on contractility [146]. In the same line, another study revealed in isolated perfused rat hearts that TRPV4 channel potentiation by the agonist GSK1016790A also had no effect on the beating rate and contractility but induced, at the in vivo level, circulatory collapse, which was most likely due to vascular leakage and tissue haemorrhage in the lungs [80]. On the other hand, our group has recently shown that perfusion of the TRPV4 agonist GSK101679A induced a significant increase in transient Ca^2+^ current influx in *trpv4^+/+^* but not in *trpv4^-/-^* mouse myocytes [8], confirming TRPV4’s involvement in modulating the left ventricular intracellular Ca^2+^ concentration. Recently, Veteto et al. explored the relationship between the TRPV4 function and Ca^2+^ handling after mechanical stimulation in aged hearts [113]. To this end, the authors explored the effect of the left ventricular preload elevation in working-heart perfused aged mice hearts and found that following the initial Frank–Starling response, these hearts exhibited a secondary increase in left ventricular maximal pressure, which was absent when perfusing the TRPV4 antagonist HC067047. Interestingly, when the preload elevation was maintained for a longer period of time (20 min), the maximal left ventricular pressure declined less in HC067047-treated than in untreated aged hearts. The authors then studied Ca^2+^ handling in ventricular myocytes following uniaxial stretching and found a delayed increase in intracellular Ca^2+^ in myocytes from aged mice, ultimately leading to a contracted state, both of which were prevented by HC067047 treatment. These results suggest that TRPV4 is responsible for Ca^2+^ entry, leading to a hypercontractile response secondary to myocardial stretching in the aged heart, although when stretched was maintained, the TRPV4-mediated Ca^2+^ influx was detrimental to cardiac contractility. Further investigations have reported that TRPV4 expression increases in mouse cardiomyocytes with advancing age [11] and after in vivo ischemia–reperfusion [61]. Note that TRPV4 upregulation in aged mice was associated with increased hypoosmotic-stress-mediated contractility, enhanced cell death and increased ischemia–reperfusion injuries [11]. Finally, the inhibition of TRPV4 by HC067047 exerts a cardioprotective effect during ischemia–reperfusion, attesting the key role of TRPV4 in Ca^2+^ handling and contractility [147]. It is important to underline that TRPV4 channel can also dock with other TRP channels and little is known about the contribution of these types of channels in intracellular Ca^2+^ dynamics and their involvement in the myocyte contractility. In this context, detailed interplay between TRPV4 channels and various α-subunits requires additional research in cardiomyocytes.

Collectively, these data suggest that the TRPV4 channel represents an attractive ion channel target to prevent Ca^2+^ overload and cardiac contractility dysfunction [10,55,61,148]. Indeed, TRPV4 seems to contribute to Ca^2+^ homeostasis regulation under physiological conditions without impacting the contractility, whereas it may be more deleterious during aging or under pathological conditions and may lead to Ca^2+^ overload and altered contractility. Further investigations are undoubtedly needed to fully understand TRPV4’s role in cardiac contractility in human cardiovascular diseases.

### 4.4. Modulation of Vascular Tone

#### 4.4.1. TRPV4 and Vasodilation

TRPV4 is expressed in ECs, where its potentiation results in vasodilatation [149,150]. TRPV4 is sensitive to shear stress [151] and is able to be activated by blood flow and induce Ca^2+^ entry when expressed in the endothelium. In ECs, Ca^2+^ entry activates the endothelial nitric oxide synthase (eNOS) pathway and the production of nitric oxide (NO), which can, in turn, diffuse to smooth muscle cells (SMCs). In SMCs, NO activates cyclic GMP/PKG signaling, inducing vasodilatation [150], leading to endothelium-derived factors release and causing SMC hyperpolarization and vasodilatation [150]. Furthermore, Ca^2+^ entry activates in parallel intermediate (IK)- and small (SK)-conductance Ca^2+^-sensitive K^+^ channels in ECs, leading to the hyperpolarization of ECs, and then the hyperpolarization of SMCs via myoendothelial gap junctions [150]. The genetic deletion or pharmacological blockade of TRPV4 channels inhibits the NO- and endothelium-derived hyperpolarizing factor (EDHF)-dependent relaxation of mouse small mesenteric arteries in response to flow [100]. Accordingly, the transfection of TRPV4 channels in HEK293 cells confers them sensitivity to flow and induces shear-stress-dependent Ca^2+^ entry [100], confirming the important role of TRPV4 in sensing shear stress. TRPV4-transfected HEK-293T cells are also sensitive to cell confluence, modulating TRPV4’s response to hypoxia at high cell densities, suggesting another important feature of TRPV4 activation, i.e., hypoxia [151]. Indeed, in vessels, hypoxia induces vasodilatation in the systemic circulation, whereas it induces vasoconstriction in the pulmonary circulation. TRPV4’s potentiation with its agonist GSK1016790A causes the endothelium-dependent relaxation of precontracted rat pulmonary artery rings, and this relaxation is inhibited in de-endothelized vessels. The authors of this study suggested that both NO and EDHF contribute to GSK1016790A-induced relaxation [120]. Ottolini et al. found that TRPV4 channels in mouse ECs colocalize with IK/SK channels in mesenteric arteries but not in pulmonary arteries, which explains that TRPV4 sparklets activate IK/SK channels in mesenteric arteries but not in pulmonary arteries, where ECs TRPV4 preferentially activate eNOS [125]. Additionally, flow-induced vasorelaxation in human coronary microvessels requires endothelial TRPV4 activation, which increases mitochondrial ROS production in ECs and induces ROS-dependent vasodilation [121,152].

#### 4.4.2. TRPV4 and Vasoconstriction

TRPV4 is also expressed in SMCs, where it participates in vascular contraction, cell migration and proliferation [124,128]. TRPV4 activation in pulmonary artery SMCs results in Ca^2+^ entry and Ca^2+^ release from the SR through the activation of ryanodine receptors [71]. This TRPV4-mediated Ca^2+^ elevation leading to SMC contraction [71,87] can be triggered by serotonin [87,126] or mechanical stimuli such as flow shear stress [127]. The dysregulation of TRPV4 expression between ECs and SMCs could lead to an impaired balance between vasorelaxation and vasoconstriction and result in an altered myogenic tone [71,122,153].

#### 4.4.3. TRPV4 and Mechanosensitivity

In vascular physiology, cells undergo mechanical stimulation induced by blood pressure, shear stress, stretching or parietal tension, which can promote TRPV4 opening. In accordance, TRPV4 was shown to be involved in the regulation of myogenic tone. In PAECs, TRPV4 can be activated by shear stress, leading to vasodilation [125]. It was also shown that TRPV4 could be activated downstream of Piezo1 in ECs, where shear stress resulted in an elevation of the intracellular Ca^2+^ concentration [154]. The elevation of intracellular Ca^2+^ due to Piezo1 was transient, whereas the TRPV4-induced Ca^2+^ response was sustained, resulting in the modification of adherent junctions or actin cytoskeleton remodeling [154]. In SMCs, intraluminal blood pressure can activate TRPV4, whose subsequent Ca^2+^ entry triggers contraction, migration and proliferation phenomena [71,87,126,127,155]. TRPV4 activity can also be regulated by membrane stiffness, as it was shown that cyclic stretching could lead to a lower cholesterol/phosphatidylcholine ratio in membranes and that cholesterol modulates TRPV4 activation to GSK1016790A or stretching [156]. An increase in mechanical stress in the vessels could lead to a dysregulation of TRPV4 signaling and to impaired physiological responses such as constriction, proliferation or migration, which are features of pulmonary hypertension.

## 5. Pathological Implications of TRPV4 Channels

### 5.1. Expression Remodeling under Pathological Condition

TRPV4 expression increases under certain pathophysiological conditions, such as aging [11], pressure overload [157,158], ischemia–reperfusion [55,61], increased membrane tension [113,114] and pericarditis in rats and patients with atrial fibrillation [51]. In this context, it seems to be important to investigate the TRPV4 expression profile during life but also during the development of cardiac diseases.

### 5.2. Arrhythmias

Cardiac arrhythmias refer to abnormal heart rhythms or significant irregularities in the electrical signals, which may alter the cardiac function or lead to sudden cardiac death. Atrial fibrillation is a common supraventricular arrhythmia characterized by rapid and irregular electrical activity in the upper chambers of the heart (atria). This arrhythmia can have various etiologies, including structural heart abnormalities, hypertension, heart valve disorders, coronary artery disease and thyroid dysfunction [159], and is associated, in part, with Ca^2+^ handling defects (connell et al.). entricular fibrillation is a life-threatening arrhythmia that occurs in the lower chambers of the heart (ventricles), causing electrical storms that can lead to torsades de pointes and sudden cardiac death (Solomon et al., Ludhwani et al.). Ventricular fibrillation is typically a consequence of an underlying heart disease, such as coronary artery disease, MI, cardiomyopathy; a primary electrical disorder; or electrolyte imbalances (Antzelevitch et al.) Unfortunately, current antiarrhythmic drugs for the treatment of atrial and ventricular fibrillation are not sufficiently specific and effective and are most of the time associated with both intra and extra-cardiac effects, which may, in turn, offset their therapeutic benefits. In this context, a deeper understanding of the maintenance and evolution of arrhythmia phenotypes may help to find adaptive therapies for cardiac patients. To date, the current therapeutic strategies use pharmacological drugs to target ion channels or limit Ca^2+^ overload and catheter-based ablation approaches. Among these channels, the TRP channel family is under the spotlight because of their biophysical properties and more specifically their permeability to Ca^2+^ [3].

Since the TRPV4 channels are expressed in native cardiac cells, including atrial cardiomyocytes, their implication in atrial fibrillation was evaluated [51]. In a model of a rat sterile pericarditis-related atrial fibrillation phenotype, the TRPV4 expression level markedly increased within the atrial tissue over 2 weeks after surgery compared to the sham condition. This result is interesting because it suggests that TRPV4 channels can be directly linked to the atrial fibrillation phenotype in the early phase of its development. The TRPV4 agonist GK1016790A perfusion on atrial myocytes increased both the action potential duration and intracellular Ca^2+^ levels, whereas the TRPV4 channel inhibition by GSK2193874 had an opposite effect. Interestingly, the authors have shown in vivo that the blockade of TRPV4 limited abnormal electrophysiological changes, protected the heart against cardiac fibrosis and inflammation and decreased the vulnerability to atrial fibrillation without explaining the accurate mechanism of this positive effect [51,160]. In this context, TRPV4 may constitute an interesting therapeutic target to treat atrial and ventricular fibrillation. Additional research is needed to identify TRPV4’s involvement in human arrhythmias

At the left ventricular myocyte level, another recent investigation pointed out the importance of considering aging and its potential negative effects on Ca^2+^ handling, the resting membrane potential and the risk of developing ventricular arrhythmia after ischemia–reperfusion. Because theTRPV4 channel is upregulated in cardiomyocytes of aged mice [11], the perfusion of the TRPV4 antagonist HC067047 after ischemia–reperfusion reduces the rate of pro-arrhythmic diastolic Ca^2+^ signaling, maintains the resting membrane potential and decreases the ventricular arrhythmia score [161]. Therefore, TRPV4 blockade may also constitute a promising therapeutic option to limit the occurrence of arrhythmogenic events for aged populations following ischemia–reperfusion and MI.

It is important to underline that the precise role of TRPV4 channels in different types of cardiac arrhythmia and their potential as therapeutic targets is still an area of active investigation. Further research is needed to fully understand the mechanisms by which TRPV4 channels contribute to arrhythmogenesis and to develop safe and effective strategies to modulate them.

### 5.3. Cardiac Remodeling and Fibrosis

There is some evidence that TRPV4 may be involved in cardiac pathological remodeling and could, thus, be an interesting therapeutic target in these contexts. In a recent study, TRPV4 channel expression was found to be significantly increased in the ventricles of mice with left ventricular pressure overload-induced hypertrophy and in failing human ventricles [158]. Interestingly, the pressure overload resulted in reduced cardiac hypertrophy, cardiac dysfunction, fibrosis and inflammation in TRPV4 knockout mice compared to wild-type animals. Moreover, treatment with GSK2193874 (TRPV4 antagonist) prevented the pressure-overload-induced remodeling and dysfunction, further confirming TRPV4’s involvement. In the same study, the authors showed that the increase in CaMKII phosphorylation found in the pressure-overloaded left ventricle in wild-type mice was absent in TRPV4 knockout mice. They suggested that this TRPV4-related increase in CaMKII phosphorylation leads to NF-κB phosphorylation and NLRP3 activation, which both contribute to the pro-inflammatory remodeling found in these pressure-overloaded hearts. Therefore, inhibiting TRPV4 appears promising to limit CaMKII phosphorylation and its multiple consequences and inflammation in cardiac hypertrophy. Despite the convincing evidence in animal models, it must be noted that TRPV4 expression regulation in human heart failure remains uncertain, with some studies showing increased expression levels [158], while others found no significant regulation [157]. Such discrepancies may be related to the heart failure etiology, as the first study [158] selected patients with dilated cardiomyopathy only, while the second [157] used a broader range of heart failure etiologies. TRPV4’s expression and function in human heart failure will require further investigations. The role of TRPV4 in cardiac remodeling has also been studied in the context of MI and subsequent fibrosis through channels expressed in CFs.

CFs originate from mesenchymal stem cells and represent two-thirds of the cardiac cell population [162]. They contribute to maintaining the structural, biochemical, mechanical and electrical properties of the healthy myocardium and constitute the main source of extracellular matrix (ECM) protein production (collagen types I and III and metalloproteases) [163]. Fibrosis is known to be related to an expansion of the cardiac interstitial space due to an accumulation of ECM proteins. During this phenomenon, CFs proliferate, migrate and differentiate into myofibroblasts [164] under the effect of a variety of factors, including transforming growth factor β1 (TGF-β1), tumor necrosis factor-α (TNFα), angiotensin II (Ang II), platelet-derived growth factor (PDGF), endothelin 1 (ET-1) and mechanical (stretching and stiffness) factors [115,165]. Interestingly, these changes in the heart eventually lead to an increased matrix stiffness, structural damaging effect and electrical disorders [6] and predispose it to diastolic dysfunction [166,167]. Moreover, intracellular Ca^2+^ signaling, which is involved in fibroblast proliferation and differentiation, is mainly activated in CFs through TRP channels, including TRPV4 channels [7,51,90,116,168]. Indeed, the accumulating evidence hints that the TRPV4 mechanosensor is an important factor in the progression of fibrosis or prevention of fibroproliferative disorders in several organs such as the skin, lungs [169], liver, kidneys, brain, blood vessels and heart [7]. The role of TRPV4 mechanotransduction in cardiac fibrosis was demonstrated using a *trpv4^-/-^* mouse model [81]. The results suggested that the TRPV4 deletion preserves cardiac function and reduces fibrosis in *trpv4^-/-^* mice but not in WT mice after transverse aortic constriction and MI [168]. Interestingly, the *trpv4^-/-^* mice displayed a marked decrease in fibrosis, as well as a decrease in the pro-fibrotic gene expression levels, including for col1A2, α-SMA, NFAT, TGF-β1 and MRTF, a mechanosensitive transcription factor. Conversely, the WT mice subjected to MI or congestive heart failure showed an upregulation of TRPV4 channels and profibrotic genes compared to the untreated WT mice, which led to increased intracellular Ca^2+^ levels and amplified the pathological fibrosis process. These findings demonstrated that TRPV4 is functional in mouse CFs and directly related to the fibrosis process insofar as its genetic deletion preserves cardiac function and protects the heart against adverse fibrosis effects [168]. In line with these results, several in vitro investigations of rat or human CFs have shown a rapid but sustained increase in Ca^2+^ influx in response to TRPV4 agonist perfusion (4αPDD [116] or GSK1016790A [82]), which promotes the fibroblasts’ differentiation into myofibroblasts. Consistently, the blockade of TRPV4 by different antagonists (AB159908 [81], RuR [116] or GSK2193874 [51]) or TRPV4 short-interference RNA (TRPV4-siRNA) [116] inhibits TGF-β-induced fibroblast differentiation [81], abolishes Ca^2+^ conductance [116] or decreases the Ca^2+^ influx level [51,115]. Recently, and in agreement with the previous investigations, treatment with the pro-fibrotic cytokine TGF-β increased the TRPV4 channel expression in human ventricular CFs [82]. Simultaneously, the authors noted both an increase in a fibrosis biomarker (the plasminogen activator inhibitor-1) and an increase in a fibroblast differentiation biomarker (the α-smooth muscle actin, α-SMA) [82]. In addition, and via the MAPK/ERK pathway, TRPV4 channel stimulation by its specific agonist (GSK1016790A) triggers the CFs’ transformation into myofibroblasts, while its antagonist (RN-9893) combined with a Ca^2+^ chelating agent (BAPTA-AM) reduces it [82]. These different points are important because they show that the Ca^2+^ permeability of TRPV4 channels is an essential component of human ventricular CF differentiation. A recent investigation demonstrated that theTRPV4 channel expression level was higher in CFs isolated from Sprague–Dawley neonatal diabetic rats and from Sprague–Dawley adult rats cultured in glucose-rich medium. Interestingly, the perfusion of the TRPV4 channel agonist HC067047 significantly abolished these CF changes. It has also been reported that HC067047 decreases collagen I synthesis and suppresses the presence of the growth factor TGF-β necessary for the differentiation of CFs into myofibroblasts. In this context, different approaches could be used directly or indirectly to manage the TRPV4 channel function and imagine new therapeutic strategies to limit fibrosis within cardiac diabetic patients [90] or in various cardiac pathologies involving many different patient populations. Finally, the link between fibrosis and the increased risk for atrial [51,170,171,172] or ventricular [115] arrhythmias has been reported in several cardiac diseases [167,173]. In this context, the anti-fibrosis approach via TRPV4 inhibition constitutes a promising therapeutic way to limit adverse outcomes and prevent arrhythmias in the heart.

Taken together, these findings are promising and identify TRPV4 as a potential therapeutic target to attenuate cardiac fibrosis, cardiac dysfunction and arrhythmias in heart failure and myocardial infarction. However, further work in human samples and patients is required to confirm these results and ensure translation to the clinic.

### 5.4. TRPV4 Channelopathies

Numerous members of the transient receptor potential channel family (TRPA1, TRPC6, TRPM1, 2, 3, 4, 6, 7, TRPML1 and TRPV3, 4) have been described as being implicated in hereditary channelopathies. In several cases, mutations disrupt ion channel function and are causal for the disease pathogenesis [174,175,176]. To date, no *trpv4* mutation has been uncovered in primary cardiac diseases and electrical disorders. However, mutations carried by the *trpv4* gene are directly associated to neurodegenerative disorders (Charcot–Marie–Tooth disease type 2C, scapuloperoneal spinal muscular atrophy, distal spinal motor neuropathy, distal spinal muscular atrophy) [177,178,179,180,181] and various skeletal displasisas ranging from mild autosomal dominant brachyolmia to severe metatropic dysplasia [17,182,183]. Moreover, it is important to underline that there is a crosstalk between both neuropathy and skeletal dysplasia, as some TRPV4 mutations, such as A217S, E278K, V620I and P799R, are responsible for linking the two phenotypes [182,184]. Interestingly, these pathologies are often associated with cardiac dysfunction and arrhythmias However, a direct link between mutated TRPV4 channels and the cardiac phenotype remains to be demonstrated.

## 6. Conclusions and Perspectives

The evidence for a crucial role of the TRPV4 channel has emerged from a large range of experiments in which its inhibition preserves the physiological intracellular Ca^2+^ dynamics and protects the heart form several dysfunctions, including pathological cardiac disorders (Figure 2), HF and arrhythmia. Unfortunately, the molecular mechanism involving the transition from a healthy heart towards a cardiac pathology is poorly understood. The use of specific TRPV4 modulators combined with transgenic animal models has given valuable information about TRPV4’s involvement in physiological or pathological cardiovascular remodeling. Note that the recent availability of the crystal structure of the TRPV4 channel will facilitate drug development to counteract diseases, including potentially pathological cardiac phenotypes. Furthermore, various genetic diseases involving abnormal TRPV4 channel function have highlighted the key role of this ion channel in a broad spectrum of cellular processes. Surprisingly, no *trpv4* mutation was found in patients with cardiac diseases so far. It would be relevant to further investigate the role of TRPV4 in the elderly population, given the increase in its expression and the risk of cardiac diseases (ischemia–reperfusion lesion, MI, pressure overload, arrhythmia) found in this population. Future work in this promising field may help to better understand the function and regulation of this channel and identify its interactions with other channels and the implications for cardiac physiological and pathological processes.

## Figures and Tables

**Figure 1 cells-12-01654-f001:**
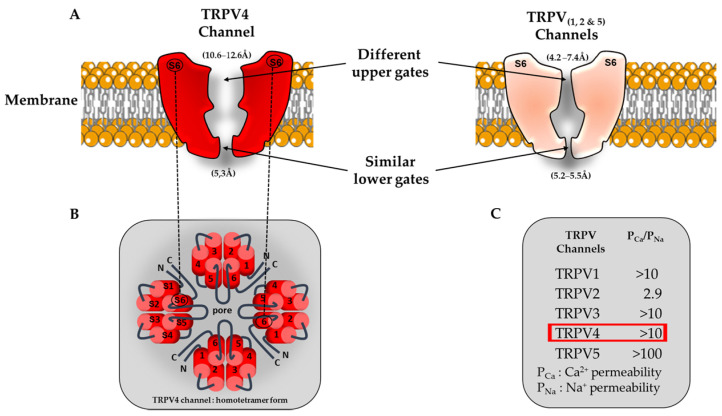
Schematic view of TRPV channel structures and associated ion permeation pathway. (**A**) Comparison of TRPV pore structures. For clarity, only two opposing subunits are shown (segment S6). Interestingly the TRPV4 upper gate is larger compared to the other members of the TRPV family (TRPV1, TRPV2 and TRPV5). (**B**) Homotetrameric structure of TRPV4. (**C**) Functional characteristics of TRPV channels (permeability ratio P_Ca_/P_Na_). The red square refers to the Ca^2+^ permeability of the TRPV4 channel.

**Figure 2 cells-12-01654-f002:**
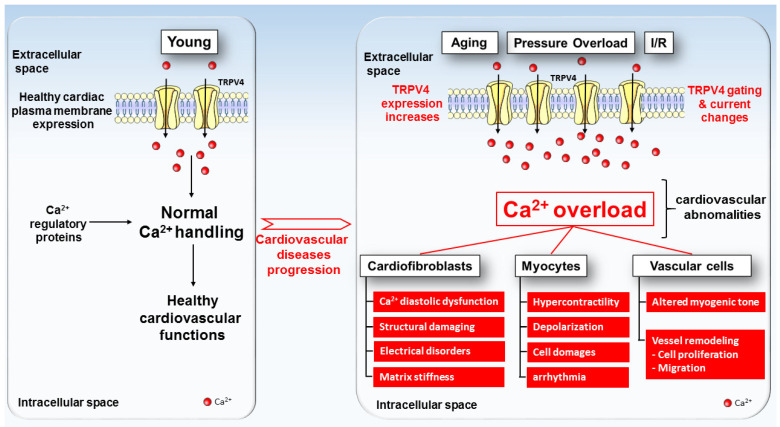
TRPV4 expression and function model in healthy (**left**) and pathologic (**right**) cardiac cells. Under physiological conditions, TRPV4 channels (yellow channels) present a low expression level in the plasma membranes of cardiac cells. With aging, pressure overload, ischemia–reperfusion and others deleterious factors, TRPV4 exhibits increases in both expression and activity in the plasma membranes of cardiac cells, which tend towards the progression of cardiovascular diseases.

**Table 3 cells-12-01654-t003:** Comparison of 129/SvJ *trpv4*^-/-^ andC57bl/6J *trpv4^-/-^*. Ref.: references.

		Ref.		Ref.
**Outcomes**	**129/SvJ *trpv4*** ** ^-/-^ **	[96]	**C57bl/6J *trpv4*** ** ^-/-^ **	[97]
Generation method	129/SvJ strain via a cassette insertion mutagenesis of exon 5		C57bl/6J strain with a Cre-lox-mediated excision of exon 12	
Cardio-vacular phenotypes	Impaired vasorelaxation, endothelial calcium response, systemic tonicity	[57,98,99,100]	Impaired vasorelaxation	[57,101,102]
	Altered flow-induced vasodilatation	[103]	Loss of shear-stress-induced vasodilation	[104]
	-		Cardiac electrophysiological changes	[8]
	-		Absence of VGIC remodeling (Na^+^, Ca^2+^ and K^+^ VGIC) in the left ventricle	[8]
Extracardiac phenotypes	Viable and fertile		Viable and fertile	[97]
	Normal appearance, growth, size and temperature and no obvious behavioral (including drinking) abnormalities	[105]	Tendency to a lower body weight	[8]
	Reduced response to harmful stimuli caused by pressure	[96]	Reductions in water intake and serum osmolality changes	[97]
	Intact heat detection but abnormal sensory phenotype	[96]	Reduced response to noxious mechanical stimuli and impaired response to mechanical stimulation	[97]
	Altered hearing	[106]	Intact thermal sensing	[97]
	Inability to thermoregulate	[99]	Loss of the permeability response in the lungs, alveolar barrier	[107]
	Deficits in renal tubular K^+^ secretion	[108]	Proximal tubule defect	[109]
	Increased bone mass, decreased osteoclast differentiation	[110]	Blood metabolite changes	[111]
	-		Increased bladder capacity	[88]

**Table 4 cells-12-01654-t004:** TRPV4 channel expression and demonstrated function in cardiovascular system.

TRPV4	Atrium	Ventricles	Fibroblasts	Endothelial Cells	Smooth Muscle Cells
mRNA	-	Mouse [61], neo-rat [10,55], rat [55]	Mouse [117], rat [116]-, human [82,117]	Mouse [100], rat [118,120], human [119,121]	Rat [120,122,123,124]
Protein	Rat [51]	Mouse [8,11,61], neo-rat [10,55], rat [55]	Rat [81], neo-rat [90], human [82,117]	Mouse [100,125], rat [118,120], human [119,121]	Rat [71,120,122,124,126], human [126,127]
Function	Rat [51]	Mouse [8,11,51,61,116], neo-rat [10], rat [51]	Rat [51,81,90], human [82,117]	Mouse [100,125], rat [118,121], human [119,121]	Mouse [83], rat [71,120,122,126,128], human [126,127]

## Data Availability

Not applicable.

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
