# Peer review of "Pathophysiological Roles of the TRPV4 Channel in the Heart"

_cells, 2023, doi:10.3390/cells12121654_

Round 1

Reviewer 1 Report

This review provides an overview of the current literature on TRPV4 channels in the cardiovascular system and with an emphasis on TRPV4 function in cardiomyocytes. The review is well-written, failry complete and comprehensive. Well structured tables provide an efficient link between specific data and the literature. 

I only have some minor suggestions:

  • Could the authors also use the available online databases of human (single cell) transcriptomic and proteomic data to underline the expression of TRPV4 in human cardiomyocytes in healthy and failing heart. 
  • After reading this review it is still unclear to me what the authors' view is on  how TRPV4 is/would be activated in a physiological setting in cardiomyocytes. Could they elaborate on this.
  • To date, extensive data shows that human TRPV4 mutations (both gain and loss of function mutations described) cause diseases such as CMT or bone dysplasia. The authors mention that there is no evidence yet for mutations in humans with cardiac disease. But could they elaborate the reverse: to what extent are cardiac abnormalities part of the disease phenotype in patients with TRPV4 mutations? 
  • Considering that virtually all available data originates from mouse work, I woould suggest to somewhat downtone statements about the "clinical significance" of TRPV4 as a drug target (line 489 among others).

the quality of English usage is good. 

Author Response

Dear Colleagues,

First of all, we would like to thank the reviewers and editor for their attention in the reading of the manuscript. We appreciated their valuable suggestions and their encouraging feedbacks. As suggested, new elements were added in the new version.

Below is our response to the reviewers.

Comments of the reviewers are in blue. Our answers are in red. Paragraphs that were added in the manuscript are in green.

Reviewer 1:

This review provides an overview of the current literature on TRPV4 channels in the cardiovascular system and with an emphasis on TRPV4 function in cardiomyocytes. The review is well-written, failry complete and comprehensive. Well structured tables provide an efficient link between specific data and the literature. I only have some minor suggestions:

Could the authors also use the available online databases of human (single cell) transcriptomic and proteomic data to underline the expression of TRPV4 in human cardiomyocytes in healthy and failing heart.

We thank the reviewer for his positive evaluation and hope that he will appreciate our responses and the new version of the manuscript.

Although there are evidences for an increased TRPV4 expression in cardiac hypertrophy and failure in animal models (refs 159 et 160), expression data from patients with heart failure, including transcriptomic data hosted at public repositories are limited and show unclear results. While some studies are reporting no change in TRPV4 expression in this context, a recent study found increased TRPV4 protein levels in patients with severe dilated cardiomyopathy (ref 160). Discrepancies may be related variable heart failure etiology and severity, patient age and history.  

We have amended the manuscript to highlight this point (line 514)

Despite convincing evidences in animal models, it must be noted that TRPV4 expression regulation in human heart failure remains uncertain with some studies showing increased expression levels (ref 160) while others found no significant regulation (ref 159). Discrepancies may be related to heart failure etiology as the first study (ref 160) selected patients with dilated cardiomyopathy only while the second (ref 159) used a broader range of heart failure etiologies. TRPV4 expression and function in human heart failure will require further investigations.   

After reading this review it is still unclear to me what the authors' view is on how TRPV4 is/would be activated in a physiological setting in cardiomyocytes. Could they elaborate on this.

ECGs recorded in vivo showed a significantly shorter QTc interval in in trpv4-/- mice than trpv4+/+. In cardiomyocytes, shorter action potentials were evidenced in trpv4-/- compared to trpv4+/+ mice using the patch-clamp technique. Altogether, these results suggest that TRPV4 channels are constitutively active in cardiomyocytes under basal conditions and contribute to the ventricular action potential. This contribution will be further potentiated in the aged myocardium. Moreover, membrane stretch due to contraction-relaxation cycles are especially under conditions of elevated ventricular load is also likely to further enhance TRPV4 current.

Therefore, we added the following paragraph in the revised version to underline this point (line 310)

 Thus, TRPV4 channels are constitutively active in cardiomyocytes from young mice under basal conditions and modulate ventricular electrophysiology (ref 8). Its contribution is likely to increase with age (ref 11). Moreover, since TRPV4 is well known to be mechanosensitive (refs 86, 91, 114) membrane stretch due to contraction-relaxation cycles especially under condition of acute increases in ventricular load (e.g. during exercise) may modulate its function.

To date, extensive data shows that human TRPV4 mutations (both gain and loss of function mutations described) cause diseases such as CMT or bone dysplasia. The authors mention that there is no evidence yet for mutations in humans with cardiac disease. But could they elaborate the reverse: to what extent are cardiac abnormalities part of the disease phenotype in patients with TRPV4 mutations?

We have now added in the revised version a short paragraph entitled TRPV4 channelopathies (line 586) in which we describe the main TRPV4 mutations found in extra-cardiac diseases (neurodegenerative pathologies and skeletal dysplasia syndromes).

5.4. TRPV4 channelopathies

Numerous members of the transient receptor potential channel family (TRPA1, TRPC6, TRPM1, 2,3,4,6,7, TRPML1 and TRPV3,4) have been described as being implicated in hereditary channelopathies. In several cases, mutations in ion channel genes that    disrupt ion channel their function and are causal for the disease pathogenesis of the disease (refs 180, 181, 182). To date, no trpv4 mutation has been uncovered in primary cardiac diseases and electrical disorders. However, mutations carried by trpv4 gene, are directly associated to human diseases, which affect mainly neuro-degenerative disorders (Charcot–Marie–Tooth disease type 2C, scapuloperoneal spinal muscular atrophy, distal spinal motor neuropathy, distal spinal muscular atrophy) (refs 183, 184, 185, 186, 187) and various skeletal displasisas ranging from mild autosomal dominant brachyolmia to severe metatropic dysplasia (refs 17, 188, 189). Moreover, it is important to underline that there is a crosstalk between both neuropathy and skeletal dysplasia insofar as some TRPV4 mutations, such as A217S, E278K, V620I and P799R, are responsible for linking of the 2 phenotypes (refs 188, 190). Interestingly, these pathologies are often associated with cardiac dysfunction and arrhythmias However, a direct link between mutated TRPV4 channels and the cardiac phenotype remains to be demonstrated.

Considering that virtually all available data originates from mouse work, I would suggest to somewhat downtone statements about the "clinical significance" of TRPV4 as a drug target (line 489 among others).

We thank the reviewer for this comment and we have decided to keep a balanced approach throughout the manuscript (see below an example come from the line 580)

Taken together, these findings are promising and identify TRPV4 as a potential therapeutic target to attenuate cardiac fibrosis, cardiac dysfunction and arrhythmias in heart failure and myocardial infarction. However further work in human samples and patients is required to confirm these results and ensure translation to the clinic.

Reviewer 2 Report

The manuscript by Chaigne et al, provides an excellent review of literature on TRPV4 structure and contribution in cardiovascular health and disease. It is a very nicely written review that will be very useful for different groups working in this space. I enjoyed reading it.

Comments:

-For every reports presented here, authors should provide the information on the system that has been used for those findings.

For example, line 73: “In mammalian cells, a proline-rich sequence in TRPV4 N-terminal can interact with the cytoskeleton protein PACSIN 3 (protein kinase C and casein kinase substrate in neurons 3) thereby regulating channel trafficking, sensitivity to membrane stretch [23] and preventing/reducing TRPV4 activation by heat [24].” What cell system, epithelial cells? HEK293 cells over expressing TRPV4? When you write, stretch, is it hypotonic cell swelling?

Please be precise throughout the manuscript and all the information that is provided for TRPV4, indicate what cell system and what model has been used.

-Resolution of figures is very poor. Ex. Fig 2. I cant read the text inside red boxes.

-Line 104: ‘In a model of pericarditis-related atrial fibrillation phenotype, TRPV4 channel inhibition 400 by GSK2193874 limited abnormal electrophysiological changes, cardiac fibrosis and inflammation, resulting in a reduced vulnerability to atrial fibrillation [52].” In what more? Please explain the model, then describe the finding. In every disease section, like Arrythmia, an introduction about the pathology in necessary.

-TRPV4 is mechanosensitive. This review needs a section on mechanotransduction via TRPV4 and its contribution to vascular physiology. How mechanical stress such as shear, stretch, stiffness affects trpv4 function.

A moderate proof reading will be useful.

Author Response

Dear Colleagues,

First of all, we would like to thank the reviewers and editor for their attention in the reading of the manuscript. We appreciated their valuable suggestions and their encouraging feedbacks. As suggested, new elements were added in the new version.

Below is our response to the reviewers.

Comments of the reviewers are in blue. Our answers are in red. Paragraphs that were added in the manuscript are in green.

Reviewer 2:

The manuscript by Chaigne et al, provides an excellent review of literature on TRPV4 structure and contribution in cardiovascular health and disease. It is a very nicely written review that will be very useful for different groups working in this space. I enjoyed reading it.

We thank the reviewer for his positive evaluation and hope that he will appreciate our responses and the new version of the manuscript.

-For every reports presented here, authors should provide the information on the system that has been used for those findings. For example, line 73: “In mammalian cells, a proline-rich sequence in TRPV4 N-terminal can interact with the cytoskeleton protein PACSIN 3 (protein kinase C and casein kinase substrate in neurons 3) thereby regulating channel trafficking, sensitivity to membrane stretch [23] and preventing/reducing TRPV4 activation by heat [24].” What cell system, epithelial cells? HEK293 cells over expressing TRPV4? When you write, stretch, is it hypotonic cell swelling?

As proposed by the reviewer, we changed the manner to present this part of the manuscript. Please find below the new version of the example (line 78):

In HEK-293 cells overexpressing TRPV4, a proline-rich sequence in TRPV4 N-terminal can interact with the cytoskeleton protein PACSIN 3 (protein kinase C and ca-sein kinase substrate in neurons 3) thereby regulating channel trafficking, and preventing/reducing TRPV4 activation by heat, cell swelling, and arachidonic acid (ref 23).

Please be precise throughout the manuscript and all the information that is provided for TRPV4, indicate what cell system and what model has been used.

Following the recommendations of the reviewer, a particular attention was paid throughout the manuscript to indicate cell systems and models used.

-Resolution of figures is very poor. Ex. Fig 2. I can’t read the text inside red boxes.

As suggested by the reviewer, we have replaced it to provide a clearer view of this figure (see below the last version.

-Line 104: ‘In a model of pericarditis-related atrial fibrillation phenotype, TRPV4 channel inhibition by GSK2193874 limited abnormal electrophysiological changes, cardiac fibrosis and inflammation, resulting in a reduced vulnerability to atrial fibrillation [52].” In what more? Please explain the model, then describe the finding. In every disease section, like Arrythmia, an introduction about the pathology in necessary.

In response to the reviewer's comments, we have expanded this part of the manuscript to include the description of cardiac pathologies (line 445). Of course and throughout the manuscript an introduction about pathology was done when necessary.

Cardiac arrhythmias refer to abnormal heart rhythms or significant irregularities in the electrical signals which may alter cardiac function or lead to sudden cardiac death. Atrial fibrillation is a common supraventricular arrhythmia characterized by rapid and irregular electrical activity in the upper chambers of the heart (atria). This arrhythmia phenotype can have various etiologies including structural heart abnormality, hypertension, heart valve disorders, coronary artery disease, thyroid dysfunction (ref 162)], and is associated, in part, to Ca2+ handling defects (ref 163). Ventricular fibrillation, is a life-threatening arrhythmia that occurs in the lower chambers of the heart (ventricles), causing electrical storms which may cause sudden cardiac death in the absence of a rapid cardioversion (refs 164, 165). Ventricular fibrillation is commonly a consequence of an underlying heart disease, such as coronary artery disease, MI, cardiomyopathy, a primary electrical disorder or electrolyte imbalances (ref 166). Unfortunately, current antiarrhythmic drugs for the treatment of atrial and ventricular fibrillation are not sufficiently specific and effective and are most of the time associated to both intra and extra-cardiac effects that may, in fine, offset their therapeutic benefits. In this context, a deeper understanding of the maintenance and the evolution of arrhythmia phenotypes may help to find adap-tive therapies for cardiac patients. To date, current therapeutic strategies use pharmacological agents to target ion channels and/or limit intracellular Ca2+ overload and catheter-based ablation approaches. Among these channels, the TRP channel family is under the spotlight because of their biophysical properties and more specifically for their permeability to Ca2+ (ref 3).

Since TRPV4 channels are expressed in native cardiac cells including atrial cardio-myocytes, their implication in atrial fibrillation was evaluated (ref 51). In a model of rat sterile pericarditis-related atrial fibrillation phenotype, TRPV4 expression level markedly increased within atrial tissue 2 weeks after surgery compared to the Sham condition. This result is interesting because it suggests that TRPV4 channel can be directly linked to atrial fibrillation phenotype in the early phase of its development. TRPV4 agonist GK1016790A perfusion on atrial myocytes increased both action potential duration and intracellular Ca2+ levels whereas the TRPV4 channel inhibition by GSK2193874 had an opposite effect. Interestingly, the authors have shown in vivo that the blockade of TRPV4 limited abnormal electrophysiological changes, protected the heart against cardiac fibrosis and inflammation and decreased the vulnerability to atrial fibrillation without explaining the accurate mechanism of this positive effect (refs 51, 167). In this context, TRPV4 may constitute an interesting therapeutic target to treat human atrial and ventricular fibrillation, additional research is needed to identify the TRPV4 involvement within this arrhythmia in human.

At the left ventricular myocyte level, another recent investigation pointed the importance to consider aging and his potential negative effect on Ca2+ handling, the resting membrane potential and the risk to develop ventricular arrhythmia after ischemia-reperfusion. Indeed and in this context and because TRPV4 channel is up-regulated in cardiomyocytes of aged mice (ref 11), the perfusion of TRPV4 antagonist HC067047 after ischemia-reperfusion, reduces incidence of pro-arrhythmic diastolic Ca2+ signaling, maintains the resting membrane potential and decreases the ventricular arrhythmia score (ref 143). Thereby, TRPV4 blockade may also constitute a promising therapeutic option interesting strategy to limit the occurrence of arrhythmogenic events for aged populations following ischemia-reperfusion and MI.

It's important to underline that the precise role of TRPV4 channels in different types of cardiac arrhythmia and their potential as therapeutic targets is still an area of active investigation. Further research is needed to fully understand the mechanisms by which TRPV4 channels contribute to arrhythmogenesis and, in fine, to develop safe and effective strategies for their modulation.

-TRPV4 is mechanosensitive. This review needs a section on mechanotransduction via TRPV4 and its contribution to vascular physiology. How mechanical stress such as shear, stretch, stiffness affects trpv4 function.

As suggested by the reviewer, we have added in the revised version a short paragraph entitled TRPV4 and mechanosensitivity (line 419) to explain the influence of the mechanical forces on its function in vascular physiology

4.4.3 TRPV4 and mechanosensitivity

In vascular physiology, cells undergo mechanical stimulations induced by blood pressure, shear stress, stretch or parietal tension which can promotes TRPV4 opening. In accordance, TRPV4 was shown to be involved in the regulation of myogenic tone. In PAECs, TRPV4 can be activated by shear stress, leading to vasodilation (ref 129). It was also shown that TRPV4 could be activated downstream of Piezo1 in ECs where shear stress resulted in an elevation of intracellular Ca2+ concentration (ref 157). The elevation of intracellular Ca2+ due to Piezo1 was transient whereas TRPV4-induced Ca2+ response was sustained resulting in modification of adherent junctions or actin cytoskeleton remodelling (ref 157). In SMCs, intraluminal blood pressure can activate TRPV4, whose subsequent Ca2+ entry triggers contraction, migration and proliferation phenomena (refs 71, 87, 127, 128, 158). TRPV4 activity can also be regulated by membrane stiffness as it was shown that cyclic stretch could lead to a lower cholesterol/phosphatidylcholine ratio in membrane and that cholesterol modulates TRPV4 activation to GSK1016790A or stretch (ref 159). An in-crease of mechanical stress in the vessels could lead to a dysregulation of TRPV4 signaling, and thus, to impaired physiological responses such as constriction, proliferation or migration which are features of pulmonary hypertension.

Round 2

Reviewer 2 Report

I don’t have any additional comments.

I don’t have any additional comments.